# Impact of Kefir on the Gut–Brain Axis: Serotonin Metabolism and Signaling in Pediatric Rats

**DOI:** 10.3390/microorganisms13112536

**Published:** 2025-11-05

**Authors:** Mehmet Gazi Boyaci, Ayhan Pektaş, Fadime Topal, Nur Önen, Bilgehan Mehmet Pektaş

**Affiliations:** 1Department of Neurosurgery, Faculty of Medicine, Afyonkarahisar Health Sciences University, 03200 Afyonkarahisar, Türkiye; mgazibyc@hotmail.com; 2Division of Pediatric Cardiology, Faculty of Medicine, Afyonkarahisar Health Sciences University, 03200 Afyonkarahisar, Türkiye; ayhan.pektas@afsu.edu.tr; 3Department of Medical Pharmacology, Faculty of Medicine, Afyonkarahisar Health Sciences University, 03200 Afyonkarahisar, Türkiye; fatmafadimetopal@gmail.com; 4Department of Pediatrics, Faculty of Medicine, Afyonkarahisar Health Sciences University, 03200 Afyonkarahisar, Türkiye; nur.onen@afsu.edu.tr

**Keywords:** Kefir, gut–brain axis, serotonin, tryptophan hydroxylase, microbiota, psychobiotics

## Abstract

Serotonin (5-hydroxytryptamine) is a key neurotransmitter involved in gastrointestinal and central nervous system functions. Given that approximately 90% of serotonin is synthesized in the gut, dietary interventions targeting the gut microbiota have emerged as promising strategies to modulate serotonin homeostasis. Kefir, a fermented milk beverage rich in probiotics and bioactive compounds, has been suggested to influence gut–brain axis signaling, yet its effects in the pediatric period remain insufficiently characterized. This study aimed to investigate the impact of kefir supplementation on serotonin biosynthesis, receptor expression, and metabolic pathways in a pediatric rat model, focusing on molecular markers across brain, jejunum, and serum tissues. Sixteen male *Wistar* rats (four weeks old) were divided into kefir and control groups. The kefir group received daily oral gavage of kefir (1 mL/100 g) for eight weeks, while controls received saline. Gene and protein expression levels of serotonergic markers (5-HT, TPH1, TPH2, SLC6A4, VMAT2, 5-HTR2B, 5-HTR3A, and 5-HTR4) were analyzed using quantitative PCR, ELISA, and Western blotting. Serotonin turnover was assessed via 5-HIAA levels. Kefir supplementation significantly increased 5-HT and TPH1 expression in both brain and jejunum tissues. In the brain, kefir elevated TPH2 and upregulated 5-HTR3A and 5-HTR2B, while reducing 5-HIAA levels, suggesting decreased serotonin degradation. In the jejunum, 5-HTR4 expression was markedly increased. Serum analyses revealed reduced TPH1/TPH2 expression but elevated 5-HTR4 levels, indicating systemic modulation of serotonergic signaling. Kefir exerts multifaceted effects on the serotonergic system in pediatric rats by enhancing serotonin biosynthesis, modulating receptor expression, and reducing serotonin turnover. These findings highlight kefir as a potential psychobiotic capable of influencing the gut–brain axis during early life, with implications for pediatric neurodevelopment and mental health. Further research, including clinical trials, is warranted to confirm its translational potential.

## 1. Introduction

Serotonin, biologically known as 5-hydroxytryptamine, is an important biogenic amine that functions both in the central nervous system and peripheral organs, playing a critical role in a wide range of physiological processes such as mood regulation, cognitive functions, sleep–wake cycle, appetite, gastrointestinal motility, vascular tone, and platelet function [1]. Serotonin, the neurotransmitter derived from the amino acid tryptophan, is synthesized in various tissues of the human body through a two-step enzymatic pathway involving tryptophan hydroxylase and aromatic L-amino acid decarboxylase. Approximately 90% of serotonin is produced in the gastrointestinal tract, particularly in enterochromaffin cells, while the remaining fraction is synthesized in the central nervous system, especially in the raphe nuclei of the brainstem [2]. In addition, serotonin is stored in platelets and released into circulation, contributing to hemostatic processes. Serotonin signaling is mediated by cell surface receptors classified into several subgroups. Most of these receptors are G protein-coupled receptors that exert their effects via intracellular second messenger systems, with the exception of the 5-HT3 receptor, which is a ligand-gated ion channel. These receptors play a pivotal role in regulating neurotransmission, muscle contraction, vasodilation, intestinal motility, and many neurological functions. In the human central nervous system, all serotonin receptor subtypes have been identified except for 5-HTR2B [3]. The biosynthesis of serotonin begins with the hydroxylation of tryptophan to 5-hydroxytryptophan by the enzyme tryptophan hydroxylase (TPH), followed by its conversion to serotonin by aromatic L-amino acid decarboxylase [4]. This biosynthetic pathway constitutes the fundamental mechanism that regulates serotonin levels in both the brain and peripheral tissues. Serotonin homeostasis is catalyzed by the monoamine oxidase (MAO) enzyme, ultimately producing 5-hydroxyindoleacetic acid (5-HIAA). As 5-HIAA is excreted in urine, it is considered the most important biomarker of serotonin degradation [5].

In recent years, it has become increasingly recognized that serotonin is not merely a neurological messenger but also closely linked to energy metabolism and nutritional processes [2]. Serotonin plays a key role in maintaining energy homeostasis. In the central nervous system, appetite is regulated through the hypothalamus, while in peripheral tissues, particularly adipose tissue and muscles, serotonin directly influences energy utilization [6]. Experimental studies have demonstrated that peripheral serotonin suppresses thermogenesis in adipose tissue, thereby reducing energy expenditure [7]. This provides a crucial mechanistic explanation for the association between elevated serotonin levels, obesity, and metabolic syndrome [8]. Moreover, serotonin has been shown to affect insulin sensitivity and modulate glucose metabolism, thereby contributing to energy balance [9]. Thus, alterations in the serotonergic system directly influence not only behavioral and neurological processes but also metabolic health. Nutrition is a major determinant of serotonin synthesis and function [10]. The dietary availability of tryptophan is a key factor in serotonin bioavailability. Tryptophan-rich foods enhance serotonergic activity in the central nervous system, while tryptophan deficiency has been associated with depression, anxiety, and appetite disorders [11]. Additionally, high-fat and high-sugar diets have been shown to negatively affect the gut microbiota, thereby impairing serotonin homeostasis, whereas fiber-rich and fermented foods promote serotonin biosynthesis. Hence, serotonin is regarded as one of the most critical biological links between nutrition and metabolic health [12].

In recent years, the gut microbiota has become one of the most extensively investigated fields in relation to serotonin homeostasis and signaling [13]. The microbiota modulates serotonin synthesis by influencing enzymes and intermediate metabolites involved in tryptophan metabolism [11]. Under physiological conditions, the gut microbiota stimulates serotonin synthesis in enterochromaffin cells, thereby contributing to signaling along the gut–brain axis. Serotonin is predominantly synthesized by enterochromaffin cells in the intestinal mucosa, and this process is modulated by gut microbiota through its metabolites that regulate TPH activity [14]. However, in cases of dysbiosis—characterized by reduced microbial diversity—this process is disrupted. Dysbiosis leads to increased intestinal permeability, heightened inflammatory responses, and imbalances in serotonin biosynthesis [15]. Consequently, both gastrointestinal and central nervous system functions are adversely affected. Particularly during childhood, disruptions in the microbiota are predicted to exert long-term effects on behavioral and neurodevelopmental processes through serotonin-mediated pathways. However, to date, no study has directly examined the correlation between gut microbiota dysregulation and the gut–brain axis in the context of serotonin [16,17].

The impact of fermented foods on gut microbiota and, consequently, serotonin homeostasis has been increasingly investigated [12]. Probiotics, defined as live microorganisms that provide health benefits to the host by modulating the gut microbiota, are abundant in kefir, which contains multiple species of *Lactobacillus*, *Lactococcus*, *Bifidobacterium*, and yeasts with probiotic properties. Kefir, a milk-derived fermented beverage, is notable for its probiotic and prebiotic properties [18]. It contains a complex microbial community composed of lactic acid bacteria, yeasts, and acetic acid bacteria [19]. In addition to antimicrobial effects [20], experimental animal studies have demonstrated its positive impact on liver dysfunction [21] and adipocyte hyperactivity [22] associated with gut microbiota imbalance, as well as on calcium metabolism [23] and tissue regeneration [24]. These diverse properties make kefir not only a food product but also a functional food. Regular consumption of kefir has been shown to enhance beneficial bacterial populations in the gut microbiota, suppress the growth of pathogenic microorganisms, reduce inflammation, and regulate intestinal permeability. Collectively, these effects contribute to more balanced serotonin homeostasis [25]. Furthermore, kefir has been suggested to increase tryptophan bioavailability, thereby supporting serotonergic activity [26].

During the pediatric period, the gut microbiota undergoes rapid development, and an individual’s microbial profile in later life is largely determined during this stage. Therefore, interventions targeting the gut–brain axis in childhood may exert long-lasting effects on neurodevelopmental processes. Regular consumption of kefir during this critical period may promote healthy gut microbiota development, thereby balancing serotonin homeostasis and supporting both gastrointestinal and neurological health. Studies conducted on pediatric rat models are of great importance for confirming this hypothesis and for understanding the potential protective effects of kefir on the gut–brain axis. In this context, investigating the effects of kefir on serotonin homeostasis and signaling in pediatric models will not only contribute to research in neurobiology and microbiota but also provide new perspectives for functional food approaches in childhood nutrition and for therapeutic strategies in psychiatric disorders.

## 2. Materials and Methods

### 2.1. Ethics, Animals and Protocols

Four-week-old male *Wistar* rats, weighing approximately 60–100 g, were maintained on a standard rodent chow diet under controlled conditions prior to the experiment. The animals were housed in a 12 h light/dark cycle at a temperature of 24–26 °C with 50–60% relative humidity. A total of 16 rats were included in the study, with 8 animals allocated to each experimental group. Approval for the study was obtained from the Ethical Animal Research Committee of Afyon Kocatepe University (AKUHADYEK 49533702–303). All experimental procedures were carried out in accordance with the Principles and Guidelines for the Care and Use of Laboratory Animals of the National Health and Medical Research Council, as well as the NIH Guide for the Care and Use of Laboratory Animals (NIH Publication No. 85–23, revised 1985). Before experimental interventions, the rats were allowed a one-week acclimatization period and subsequently divided into two groups: control and kefir. During the 8-week experimental period, both groups consumed feed and drinking water *ad libitum*. Moreover, kefir group received kefir daily by gastric gavage at a dose of 1 mL per 100 g body weight, while the control group was administered an equivalent volume of saline (1 mL/100 g) for duration of eight weeks. Kefir was obtained commercially (Danem Kefir, Isparta, Türkiye) and freshly prepared and stored at 4 °C; control animals received equal-volume saline. The weekly food and fluid intake of the rats were monitored throughout the experimental period. The standard rodent chow provided an energy content of approximately 3.1 kcal/g, while kefir supplied an energy value of 5.9 kcal per 1 mL/kg. According to the manufacturer’s reports, kefir yeast contains *Lactobacillus* (10.54 log CFU/mL), *Lactococcus* (10.62 log CFU/mL), *Yeast* (2.89 log CFU/mL), *Lactobacillus acidophilus* (8.25 log CFU/mL), and *Bifidobacterium* (7.78 log CFU/mL). More specifically, it contained *Lactobacillus reuteri*, *Lactobacillus fermentum*, *Lactobacillus parakefiri*, *Lactobacillus casei*, *Lactobacillus acidophilus*, *Lactococcus lactis*, *Lactobacillus helveticus*, *Leuconostoc mesenteroides*, *Lactobacillus bulgaricus*, *Lactobacillus kefiranofaciens*, *Kluyveromyces marxianus*, *Acetobacter pasteurianus*, *Bifidobacterium bifidum*, *Saccharomyces cerevisiae*, *Streptococcus thermophilus*, and *Kluyveromyces lactis*. At the end of the experimental period, the rats were anesthetized with xylazine (10 mg/kg) and ketamine (100 mg/kg). Thirteen-week-old Wistar rats (Starting at 4-week; plus 8 weeks after addition of diet) were used in this study, corresponding to the late adolescent or young adult stage in humans. Brain and jejunum tissues were harvested and rinsed with physiological saline before being snap-frozen in liquid nitrogen and stored at −85 °C. Intracardiac blood samples of at least 4 cc were also collected; at least 1 cc was placed in EDTA tubes for RT-PCR analysis and frozen at −85 °C; the rest was transferred to serum tubes, centrifuged at 4 °C and 10,000× *g* for 10 min to separate the supernatant, and then stored under the same conditions as the tissue samples.

### 2.2. Gene Expressions of SLC6A4, 5-HT, 5-HTR4, 5-HTR3A, 5-HTR2B, TPH1, and VMAT2 with Real—Time Polymerase Chain Reaction

Total RNA was isolated from brain–jejunum tissues and serum samples using the A.B.T.™ Blood/Tissue RNA Purification Kit (Acrobiosystems, Basel, Switzerland) following the manufacturer’s protocol with Proteinase K digestion and chloroform–isoamyl alcohol extraction. RNA purity and concentration were assessed with a NanoDrop spectrophotometer (AG Dammstrasse, Neumarkt, Germany), and 1 µg of RNA (tissue) or 50 ng (blood) was reverse transcribed using the VitaScript™ First-Strand cDNA Synthesis Kit (Vitagene, San Francisco, NC, USA). Gene expression was analyzed by quantitative real-time PCR (qPCR) on an Applied Biosystems 7500 system with 2X Magic SYBR Green Master Mix (Roche, Basel, Switzerland). Reactions were carried out in a final volume of 20 µL containing 10 ng cDNA, 10 µM primers, and nuclease-free water. Cycling conditions consisted of an initial denaturation at 95 °C for 5 min, followed by 40 cycles of 95 °C for 15 s, 60 °C for 30 s, and 72 °C for 30 s, with melt curve analysis performed to verify amplification specificity. Target genes included SLC6A4, 5-HT, 5-HTR4, 5-HTR3A, 5-HTR2B, TPH1, and VMAT2, with β-actin serving as the endogenous control (Table 1).

### 2.3. Determination of 5-HT, TPH1, TPH2, 5-HTR3-A, 5-HTR2B, and 5-HIAA by ELISA Assay

Tissue total protein levels were measured using the Thermo Scientific Coomassie Plus Bradford Assay Kit (Prod #23236), with absorbance read at 595 nm on a Chromate 4300 microplate reader (Awareness Technology, Palm City, FL, USA). Standard curves were generated with BSA dilutions and data were calculated using point-to-point analysis. In addition, serum, brain, and jejunum samples were analyzed for 5-HT, TP, TPH1/2, 5-HTR3A/2B, and 5-HIAA using commercially available rat ELISA kits (Bioassay Technology Laboratory, Shanghai, China). Measurements were performed at 450 nm, and concentrations were determined using linear regression analysis with correlation coefficients (R^2^) above 0.99 for all assays.

### 2.4. Evaulation of 5-HT and TPH1 Expressions by Western Blot

Western blot analyses were performed to assess 5-HT, TPH1, and β-actin protein expression in brain and jejunum tissue samples. Protein concentrations were determined using the Qubit Protein Assay Kit (Thermo Fisher Scientific, Waltham, MA, USA), and equal amounts (50 µg per lane) were loaded onto Bolt™ 4–12% Bis-Tris Plus gels for electrophoresis. Proteins were transferred to membranes using the iBlot™ 2 Dry Blotting System and subsequently blocked before antibody incubation. Primary antibodies against β-actin, TPH1, and 5-HT (ABclonal Technology, Woburn, MA, USA) and HRP-conjugated goat anti-rabbit IgG secondary antibody were applied using the iBind™ Flex automated system. Detection was achieved with enhanced chemiluminescence (ECL) and signals were visualized using the Genbox imagER Fx-CFx imaging system.

### 2.5. Statistical Analyses

All data are presented as mean ± standard error of the mean (SEM) throughout the study. Statistical comparisons were performed using Student’s *t*-test. A *p*-value of less than 0.05 was considered indicative of statistical significance. Relative changes are expressed as percentage differences compared to the Control. Expression data other than ELISA were normalized with corresponding beta-actin.

## 3. Results

### 3.1. Initial and the Final Body Weights

As shown in Figure 1, body weight gain of male *Wistar* rats in the Control and Kefir groups following an 8-week dietary intervention is presented. During the experimental period, control rats consumed an average of 24.1 g of food and 15 mL/100 g body weight of drinking water daily, corresponding to a daily energy intake of 74.7 kcal. Rats in the kefir group consumed 24.8 g of food and 1 mL/kg of kefir daily, resulting in a higher total daily energy intake of 82.8 kcal. Although rats in the Kefir group received a higher daily caloric intake their weight gain did not differ significantly from that of the Control group.

### 3.2. mRNA Expressions of SLC6A4, 5-HT, 5-HTR4, 5-HTR3A, 5-HTR2B, TPH1, and VMAT2

The results of mRNA expression analysis obtained by real-time qPCR are presented in Figure 2. In the brain, kefir consumption significantly decreased 5-HTR4 expression compared with the control group, while no significant differences were observed in the other parameters (Figure 2A). In the jejunum, kefir supplementation led to a significant reduction in serotonin (5-HT) mRNA levels whereas 5-HTR4 expression was increased. Additionally, TPH1 and VMAT2 expression levels showed a tendency to increase with kefir consumption; however, these changes did not reach statistical significance (Figure 2B). In the serum samples (Figure 2C), kefir intake decreased the expression of 5-HT, 5-HTR2B, and TPH1, while 5-HTR4 expression was elevated compared with the control group.

### 3.3. 5-HT, TPH1, TPH2, 5-HTR3-A, 5-HTR2B, and 5-HIAA Levels Detected by ELISA Kits

The expression results assessed by ELISA are shown in Figure 3. In the brain tissues, kefir consumption enhanced expression of TPH2, while 5-HTR3A and 5-HTR2B levels were also increased. Conversely, levels of 5-HIAA, a serotonin metabolite, were significantly reduced. Although 5-HT, TP, and TPH1 levels exhibited a showed a tendency toward increase elevation in the kefir group, these changes did not reach statistical significance (Figure 3A). In the jejunum, kefir supplementation resulted in a significant reduction in TP and TPH2 levels, whereas no significant alterations were observed for the other parameters. (Figure 3B) In the serum samples, kefir intake increased TP and 5-HTR2B levels, while TPH1 and TPH2 were decreased. Additionally, 5-HIAA levels showed a trend toward elevation with kefir consumption (Figure 3C).

### 3.4. Western Blot Analyses

The Western blot analysis of TPH1 and 5-HT expression in brain and jejunum tissues is presented in Figure 4. In the brain tissues, kefir consumption led to an upward trend in TPH1 expression; however, this increase did not reach statistical significance. In contrast, 5-HT levels were significantly elevated in the kefir group compared with controls (Figure 4A). In the jejunum, both TPH1 and 5-HT levels were markedly upregulated in rats that had received kefir, indicating significant enhancement compared to the control group (Figure 4B).

## 4. Discussion

The present study shows the modulatory effects of kefir on serotonin and signaling pathways within the gut–brain axis of pediatric rats, revealing a multifaceted impact across central and peripheral systems. By evaluating the results from a broad perspective, the importance of kefir consumption in serotonin-related disorders will be understood. In this study, the choice of kefir brand, dosage, and duration of dietary intervention was determined based on the rationale and methodological framework established in our previous research [23,24,27]. Despite the higher daily caloric intake observed in the kefir group compared to the control group, no significant increase in body weight was detected. Interestingly, although the baseline body weights of the animals in the kefir group were slightly higher than those in the control group, by the end of the eighth week the weights of both groups had converged. This finding is consistent with our earlier studies, in which kefir demonstrated a weight-reducing effect particularly in obese rats induced by metabolic syndrome [21,22]. Taken together, these observations suggest that kefir may exert modulatory effects on weight regulation, potentially counteracting calorie-induced weight gain through mechanisms that warrant further investigation, such as alterations in gut microbiota composition, energy metabolism, or satiety-related pathways. Although kefir supplementation markedly altered the expression of several serotonergic markers, no significant changes were detected in SLC6A4 or VMAT2 levels in brain, jejunum, or serum samples. This finding suggests that kefir’s modulatory effects on serotonin homeostasis are more closely related to biosynthetic enzymes (TPH1/2) and receptor-level regulation rather than to serotonin reuptake or vesicular storage mechanisms. SLC6A4 primarily mediates synaptic serotonin reuptake, and its expression is relatively stable under physiological conditions unless exposed to strong pharmacological interventions, such as selective serotonin reuptake inhibitors (SSRIs) [1,3]. Similarly, VMAT2 regulates presynaptic vesicular storage of monoamines, and its expression has been shown to remain stable despite dietary or microbial manipulations, with changes usually occurring only under neurotoxic or pharmacological stress [6,10]. Therefore, the absence of significant modulation in these transport systems indicates that kefir’s effects are not mediated through altering serotonin reuptake or vesicular packaging but rather through enhancing serotonin biosynthesis and modulating receptor activity. This selective pattern of influence underscores kefir’s potential to fine-tune serotonergic signaling without disturbing essential baseline mechanisms of serotonin handling, which may contribute to its safety profile as a functional food intervention.

Undoubtedly, a more comprehensive interpretation of serotonergic variability requires the simultaneous evaluation of 5-HT and TPH levels across brain, jejunum, and serum samples. Our findings indicate that following kefir consumption, serotonin mRNA expression remained unchanged in the brain but was reduced in both the jejunum and serum. In contrast, protein measurements by ELISA demonstrated an increasing trend in brain tissues, while Western blot analyses revealed a dramatic upregulation of 5-HT expression in both brain and jejunum. Furthermore, TPH2 levels were found to increase in the brain while decreasing in the jejunum and serum, whereas TPH1 levels showed a partial increase in the brain but a pronounced elevation in the jejunum. It is well established that TPH enzymes are the key regulators of serotonin biosynthesis, with TPH1 predominantly localized in peripheral tissues, particularly the intestine, and TPH2 primarily expressed in the central nervous system [1,11]. Accordingly, our results suggest that kefir modulates serotonin homeostasis in a tissue-specific manner, exerting a strong activation of intestinal serotonergic pathways through TPH1, while simultaneously enhancing central serotonergic activity via TPH2 upregulation in the brain. This dual mechanism highlights the capacity of kefir to enhance serotonin bioavailability both peripherally and centrally, thereby supporting gut–brain axis homeostasis. One possible explanation for the apparent discrepancies between mRNA expression and protein levels across tissues is the presence of post-transcriptional and post-translational regulatory mechanisms. It is well recognized that mRNA abundance does not always directly correlate with protein expression due to differences in translation efficiency, protein stability, and regulatory feedback loops [28,29]. In our study, while serotonin mRNA levels decreased in jejunum and serum, protein analyses revealed a robust increase, particularly in jejunal tissues. This suggests that kefir may enhance the translation or stability of serotonin-related proteins rather than uniformly increasing transcriptional activity. Such effects could be mediated by microbial metabolites in kefir, including which are known to influence epigenetic regulation and protein turnover [10,14]. Another factor that may contribute to these findings is the tissue-specific localization of TPH isoforms. TPH1, primarily expressed in enterochromaffin cells, showed a strong induction in the jejunum, consistent with kefir’s local impact on the gut microbiota and its metabolites. Conversely, TPH2, localized in the brain, increased centrally but declined peripherally, suggesting that kefir’s influence on central serotonin synthesis may rely more on neural pathways and precursor availability than on direct transcriptional induction in peripheral tissues. This distinction supports the hypothesis that kefir exerts region-specific effects on serotonergic regulation, possibly through gut-derived signaling molecules acting on the vagus nerve or via systemic circulation of tryptophan and its metabolites [13,16]. Furthermore, methodological differences between qPCR, ELISA, and Western blot analyses may account for some variation in observed results. While qPCR strictly measures gene expression at the transcriptional level, ELISA and Western blot provide insights into protein abundance and post-translational modifications. The combination of decreased mRNA but increased protein levels could therefore reflect a biological scenario in which kefir promotes protein accumulation despite reduced transcriptional activity, a phenomenon previously described in probiotic studies [12]. Finally, it is important to consider the developmental context. Pediatric rats exhibit heightened plasticity in both their microbiota and serotonergic systems. The jejunum, as a primary site of microbial colonization, may undergo rapid adaptive responses to kefir supplementation, explaining the dramatic upregulation of TPH1 protein despite reduced mRNA levels [13]. In contrast, the brain may be less susceptible to immediate transcriptional changes but more responsive to long-term regulatory cues, resulting in the observed increase in TPH2 protein. Together, these mechanisms illustrate that kefir does not uniformly modulate serotonin metabolism but instead shapes it through complex, tissue-specific, and multi-level regulatory processes. The levels of tryptophan, the primary substrate for serotonin, showed a partial increase in the brain, a significant decrease in the jejunum, and a significant increase in the serum following kefir consumption (Figure 3). This pattern likely reflects enhanced 5-HT synthesis in the jejunum and suggests that the body may facilitate continuous serotonin production by increasing tryptophan absorption or synthesis. Furthermore, the observed decrease in 5-HIAA levels in brain tissue supports these interpretations. Supporting this interpretation, previous studies have shown that reduced 5-HIAA levels correlate with sustained serotonergic tone and improved emotional regulation [4,10]. Moreover, probiotic interventions have been demonstrated to suppress monoamine oxidase activity, leading to reduced serotonin catabolism [16]. The fact that our study replicated this pattern in a pediatric model strengthens the translational relevance of kefir as a dietary adjunct for conditions associated with serotonergic dysfunction. Beyond biosynthesis and metabolism, kefir also induced receptor-specific changes in serotonergic signaling. Increased expression of 5-HTR4 in jejunum and serum samples, alongside elevated 5-HTR3A and 5-HTR2B receptor levels in the brain, highlight the capacity of kefir to modulate downstream receptor pathways. These receptors mediate diverse functions, including gastrointestinal motility, synaptic plasticity, and emotional behavior [1,3]. The receptor-level effects observed in this study align with previous findings where probiotics enhanced serotonergic receptor sensitivity, ultimately influencing behavior and stress responses [12,30]. Importantly, receptor modulation may represent a complementary mechanism by which kefir exerts neuromodulatory effects beyond the simple elevation of serotonin levels. The pediatric period is particularly critical for the development of both the gut microbiota and serotonergic system. Dysbiosis during this window has been linked to long-term disturbances in neurodevelopment, behavior, and metabolic regulation [15,17]. Our findings suggest that kefir supplementation during this stage may mitigate these risks by stabilizing microbial diversity and enhancing serotonergic homeostasis. This has important implications for conditions such as autism spectrum disorder, attention-deficit/hyperactivity disorder, and pediatric anxiety, all of which have been associated with microbiota–serotonin dysregulation [13,16]. By promoting microbial balance and facilitating serotonin availability, kefir could serve as a natural, non-invasive intervention to support neurodevelopmental resilience. In addition to its neurobiological effects, kefir’s influence on serotonergic pathways may intersect with its known metabolic benefits. Previous studies demonstrated that kefir protects against liver dysfunction, adipose tissue hyperplasia, and systemic inflammation, primarily through restoration of microbiota balance and modulation of metabolic signaling [21,22,24]. Serotonin itself plays a crucial role in regulating energy homeostasis, insulin sensitivity, and glucose metabolism [6,9]. Therefore, the serotonergic changes induced by kefir in our study may partially underlie its systemic metabolic benefits, further reinforcing the concept of kefir as a functional food with multi-organ impact. Several mechanistic pathways may account for the observed effects. First, microbial metabolites generated during kefir fermentation—such as short-chain fatty acids (SCFAs) and bioactive peptides—can influence both gut and brain function directly [14]. Second, kefir’s anti-inflammatory and antioxidant activities may protect against cytokine-driven tryptophan degradation through the kynurenine pathway, thereby preserving serotonin availability [8,16]. Third, kefir may enhance vagal nerve signaling by increasing intestinal serotonin release, providing a direct neural route for gut–brain communication [13]. Although peripheral serotonin does not cross the blood–brain barrier, kefir may influence central serotonergic signaling indirectly through vagal activation and modulation of tryptophan metabolism. Each of these pathways likely contributes in concert to the serotonergic modulation observed in our model. Despite the promising results, some limitations should be acknowledged. The relatively small sample size and use of only male rats may limit the generalizability of findings, as sex-specific differences in serotonergic regulation have been documented [2]. Furthermore, although molecular changes were robustly demonstrated, behavioral correlates such as anxiety, learning, and memory performance were not assessed. Previous research has shown that microbiota-targeted interventions can produce measurable behavioral changes [15,30]; thus, future studies should integrate both biochemical and behavioral endpoints. Another limitation concerns the variability in kefir composition across different sources and fermentation conditions. Microbial strains present in kefir can differ widely, potentially influencing outcomes [25]. Therefore, standardization of kefir products will be essential for clinical translation.

In conclusion, this study provides novel evidence that kefir modulates serotonin biosynthesis, receptor signaling, and degradation across central and peripheral systems in pediatric rats. By enhancing TPH1-mediated serotonin production, reducing 5-HIAA levels, and upregulating key serotonergic receptors, kefir exerts a multifaceted influence on the gut–brain axis. These findings are consistent with and extend previous reports demonstrating the role of probiotics and fermented foods in regulating serotonergic function [10,12,16]. Given the critical importance of early-life microbiota development for long-term neuropsychological outcomes, kefir emerges as a promising psychobiotic candidate for pediatric nutrition. Further research, particularly clinical trials, will be necessary to confirm these effects in humans and to determine the behavioral and therapeutic significance of kefir as a functional food in the prevention and management of serotonin-related disorders.

## Figures and Tables

**Figure 1 microorganisms-13-02536-f001:**
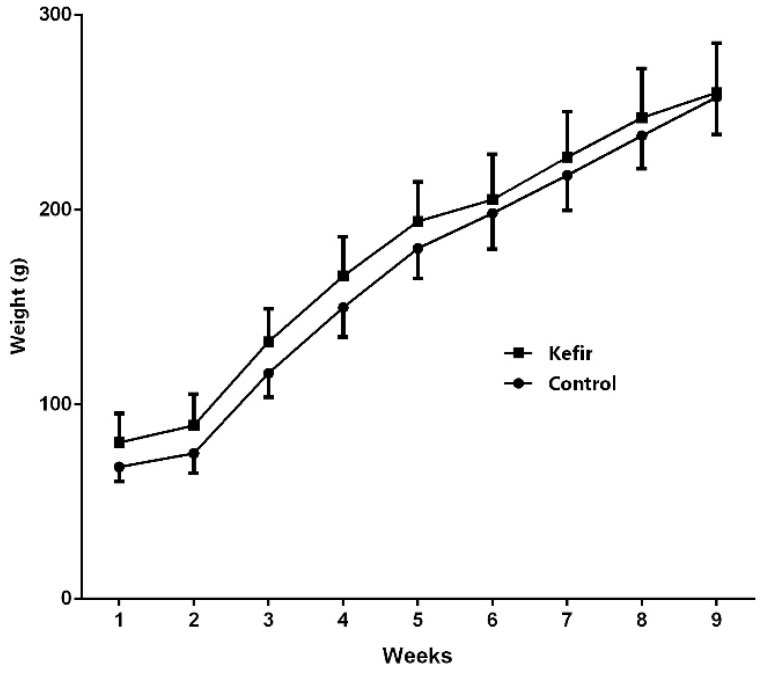
Point (1) represents the initiation of the diet, whereas point (9) denotes its endpoint. Body weight changes of *Wistar* rats over an 8-week period. The graph presents mean body weight (g) of the Control group (●) and the Kefir group (■), which received daily kefir supplementation. Each group consisted of 8 rats. Data are expressed as mean ± SEM.

**Figure 2 microorganisms-13-02536-f002:**
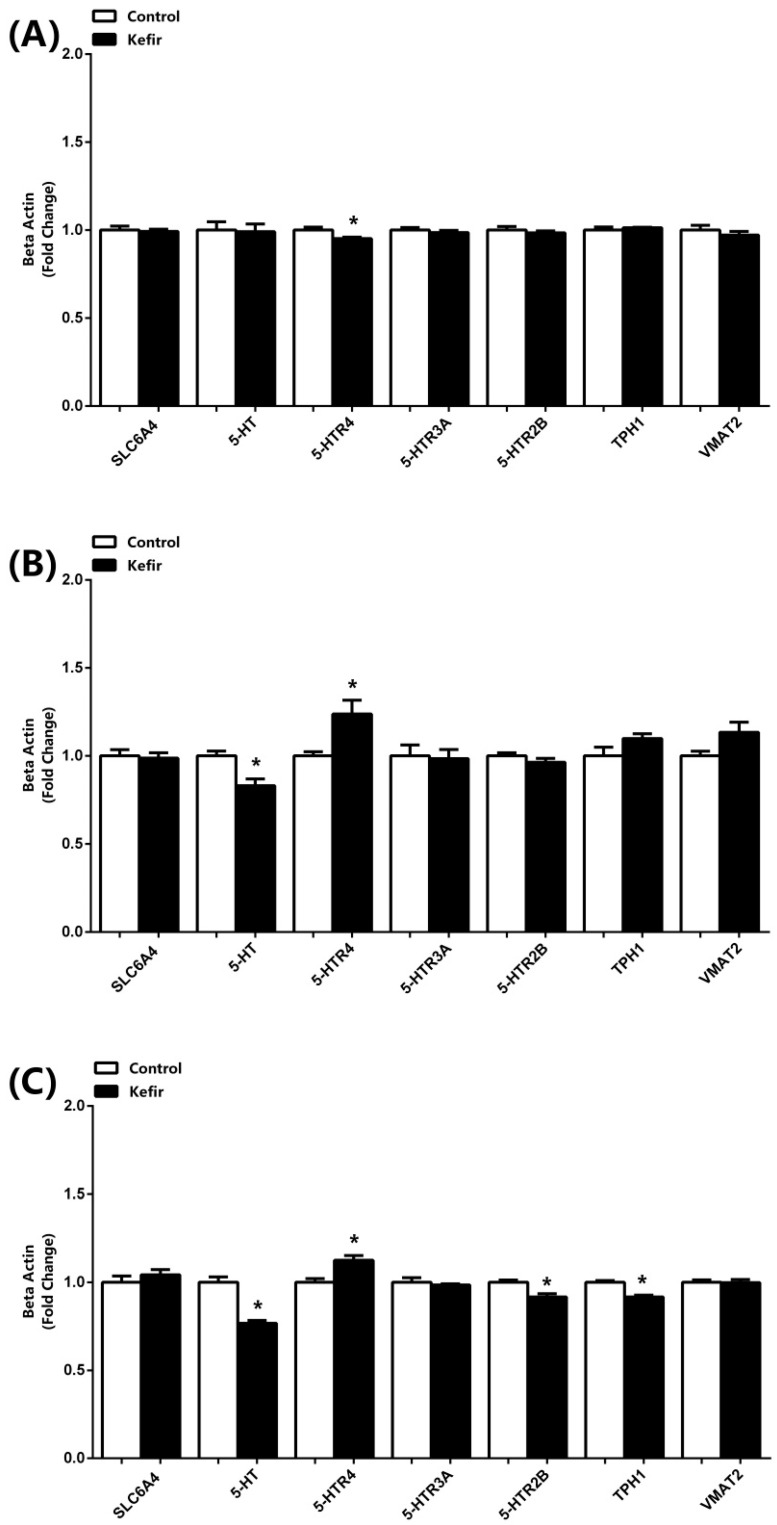
mRNA expression levels of SLC6A4, 5-HT, 5-HTR4, 5-HTR3A, 5-HTR2B, TPH1, and VMAT2 in the brain (**A**), jejunum (**B**), and serum (**C**) samples of male rats from the Control and Kefir groups. Data was normalized by β-actin and presented as fold-change relative to the Control. Each bar represents the means of at least six rats. Values are expressed as mean ± SEM, * Significantly different (*p* < 0.05) compared to Control group. 5-HT, 5-Hydroxytryptamine (Serotonin); TPH1-2, Tryptophan Hydroxylase 1-2; SLC6A4, Serotonin Transporter; VMAT2, Vesicular Monoamine Transporter 2; 5-HTR2B-3A-4, 5-Hydroxytryptamine Receptor 2B-3A-4.

**Figure 3 microorganisms-13-02536-f003:**
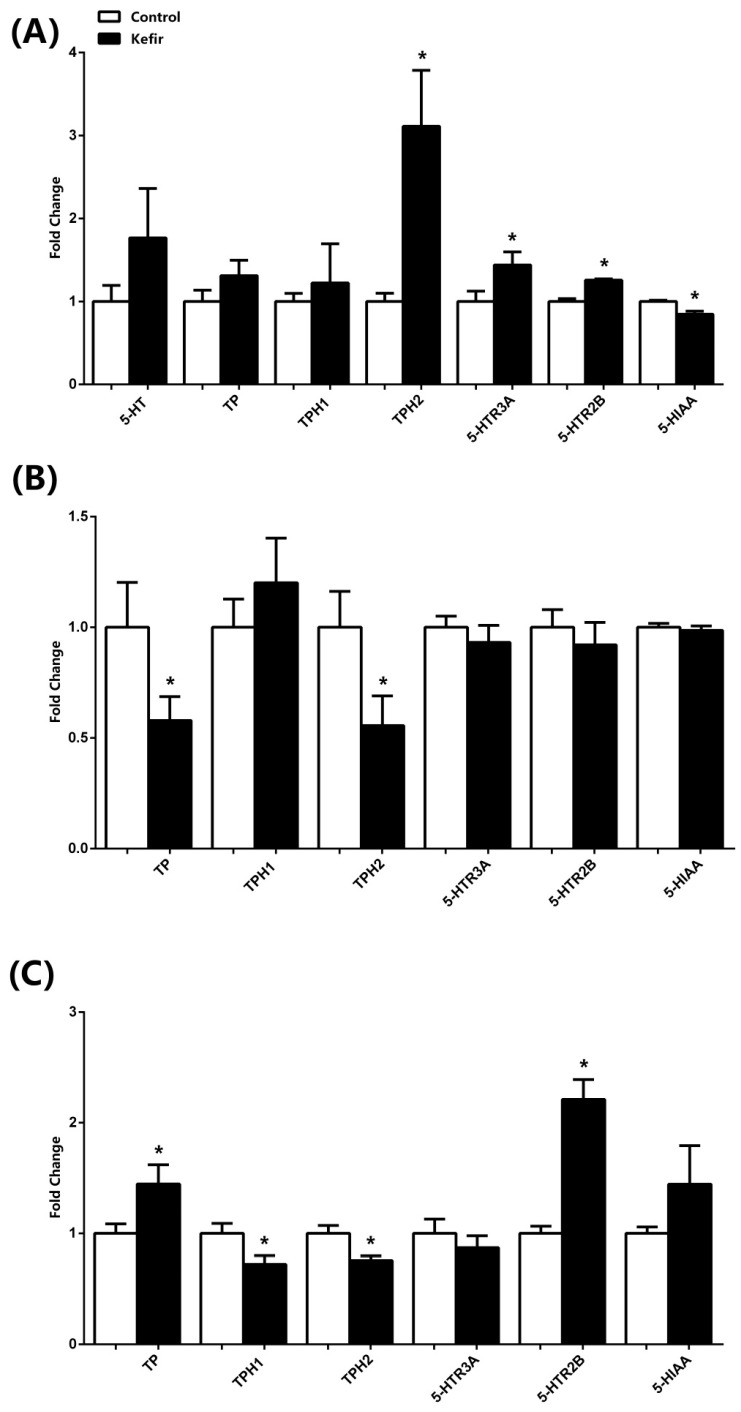
Levels of 5-HT, TPH1, TPH2, 5-HTR3A, 5-HTR2B, and 5-HIAA measured by ELISA in the brain (**A**), jejunum (**B**), and serum (**C**) samples of male rats from the Control and Kefir groups. Data are presented as fold-change relative to the Control. Each bar represents the mean of at least six rats, and values are expressed as mean ± SEM. * *p* < 0.05 compared with the Control group. 5-HT, 5-Hydroxytryptamine (Serotonin); 5-HIAA, 5-Hydroxyindoleacetic Acid; TPH1-2, Tryptophan Hydroxylase 1-2; 5-HTR2B-3A-4, 5-Hydroxytryptamine Receptor 2B-3A-4.

**Figure 4 microorganisms-13-02536-f004:**
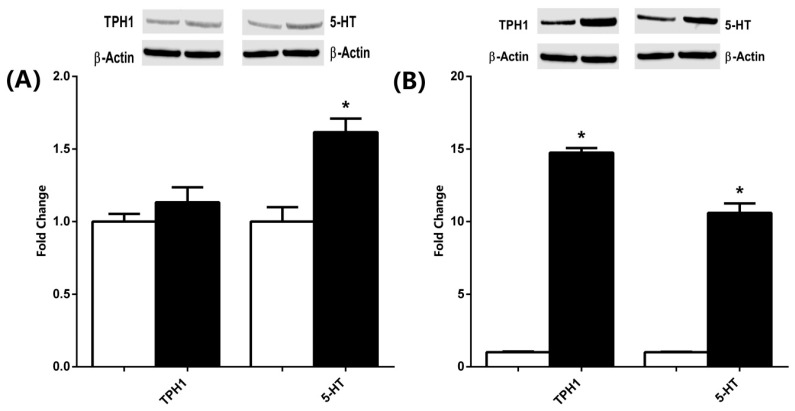
The expression levels of 5-HT and TPH1 in the brain (**A**) and jejunum (**B**) samples of male rats from the Control and Kefir groups. The levels were quantified using densitometry and normalized with by β-actin and data presented as fold-change relative to the Control. Representative Western blot images are included above the corresponding figures. Each bar represents the means of at least six rats. Values are expressed as mean ± SEM, * Significantly different (*p* < 0.05) compared to Control group. 5-HT, 5-Hydroxytryptamine (Serotonin); TPH1, Tryptophan Hydroxylase 1.

**Table 1 microorganisms-13-02536-t001:** Primer sequences of SLC6A4, 5-HT, 5-HTR4, 5-HTR3A, 5-HTR2B, TPH1, and VMAT2, with internal standard β-actin used for specifying mRNA expression by *qRT-PCR*.

Gene	Forward Primer Sequence (5′ → 3′)	Reverse Primer Sequence (3′ → 5′)	Product
** *SLC6A4* **	CCGTCATCTGCATCCCTACC	ATGTCCCCACACGGGATTTC	20
** *5-HT* **	GGACTCCTCCTCTAAGCAAGC	CACGGAAAGAAGTGGTCGGA	21
** *5-HTR4* **	GGCTCACGAGGAGATGTCTG	TAGAGGGAGGGTGGGTTCAG	20
** *5-HTR3A* **	TTGGCCTTGTTCCTTTCCGT	CGCACCCCCTTCTTGTAGTT	20
** *5-HTR2B* **	ATCTGTCAGGGGAGGGAGTC	TTTCAGAAGATGCTTGTCTGCTT	23
** *TPH1* **	TGCGACATCAACCGAGAACA	CGGATCCGTACAACAGCACT	20
** *VMAT2* **	CCATGGCCCTGAGCGATCT	CTGGTGGTCTGGATTTCCGT	19
** *β-ACTIN* **	CCAGGAGTACGATGAGTCCG	ACGCAGCTCAGTAACAGTCC	20

## Data Availability

The data that support the findings of this study are available on request from the corresponding author.

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
