# Peer review of "Impact of Kefir on the Gut–Brain Axis: Serotonin Metabolism and Signaling in Pediatric Rats"

_microorganisms, 2025, doi:10.3390/microorganisms13112536_

Round 1

Reviewer 1 Report

Comments and Suggestions for Authors

In my opinion, the main weakness of this study is the use of a commercial product without analyzing its composition to ensure the reliability of the results and conclusions. Moreover, using only eight animals per group may be insufficient, especially considering that the authors used pediatric rats and presented data such as body weight.

Author Response

We would like to thank both reviewers for their valuable comments and suggestions, which have significantly contributed to improving the quality and clarity of our manuscript. Below are our point-by-point responses to each comment.

  1. “In my opinion, the main weakness of this study is the use of a commercial product without analyzing its composition to ensure the reliability of the results and conclusions. Moreover, using only eight animals per group may be insufficient, especially considering that the authors used pediatric rats and presented data such as body weight.”

RESPONSE: Thank you for this insightful comment. We agree that characterizing the composition of probiotic products can improve the interpretability of experimental outcomes. In the present study, our primary objective was to investigate the functional serotonergic effects of kefir on the gut–brain axis rather than to conduct a compositional microbiological analysis. For this reason, we used a standardized commercial kefir product with a documented microbial profile provided by the manufacturer. The method section presents the bacteria contained in commercial kefir mixes. The primary reason for using this product, unlike other products, is its close proximity to the bacterial microbiota diversity. Compared to other products, they only offer a bacterial diversity of 2-5%. Furthermore, we have previously conducted numerous studies on the same product and published the results in prestigious journals such as Microorganisms (Aslan et al., 2023; Ekici et al., 2022a-b; Pektas et al., 2022). Nevertheless, thank you for your attention to this topic.

Moreover, thank you for this important point (n). We agree that larger cohorts can enhance statistical power and biological generalizability. Our study design was based on standard pediatric rat models commonly used in experimental microbiota research, in which sample sizes of 6–10 animals per group are broadly accepted for molecular outcome measures. Moreover, the use of pediatric animals requires ethical and logistical restrictions that limit group size, as emphasized in previous studies.

Reviewer 2 Report

Comments and Suggestions for Authors

This is an interesting and important study, but there are several formal and methodological questions to be answered and aspects to be corrected.

Major:

Abstract: L.15f: The link between synthesis in the gut and its microbiota is not clear. Define the cells producing serotonin in the gut and the link to the intestine's microbiota. L 18: define these probiotics briefly. L. 25f: Clarify abbreviations at their 1st mentioning. L.28: remove 'protein'.

Further text:

  1. 125ff: the protocols of weighing and quantitative assessment of food intake are missing. Moreover, (L. 127) why is the range of weight that high? Were the animals from different breeds? If litter size is highly different, the weight and developmental pattern as well as nutrition are different, due to different start ages of feeding on chow. Please, clarify these issues. Allocation to groups must be described. Although group weight was apparently not significantly different, it is unclear, how the allocation to groups was performed and no explanation of different means at start is explained.

Moreover, the sex of animals and distribution between groups isn't declared, nor the logistics and maintenance conditions of kefir and placebo solution.

  1. 140ff: at an age of 13 weeks rats have reached breeding age (6-12wks). The authors should relate the developmental characteristics of rats to those of humans here.
  2. 150ff: Why were different parts of the brain not isolated and snap-frozen in separate?
    Why were the colon and caecum not harvested and analyzed for microbial analysis. Characterize, why only the jejunum of the small intestine was harvested. Moreover, define volumes of harvested blood. How was EDTA blood treated and analyzed? Clarify that whole blood was used for RT-PCR.
  3. 156 and elsewhere: all abbreviations must be defined at their first use! Include it as a table or abbreviation list with cross reference.
  4. 194ff (Statistics): clarify, whether data were analyzed for normal distribution. If not the case, present data as medians and interquartile ranges, and use non-parametric testing. Indicate, whether data were corrected for multiple testing.

Figures: Any abbreviation in a figure must be explained in its legend so that a figure is self-explanatory.

  1. 232: a non-significant change isn't a change, and cannot be a 'marked elevation'!
  2. 233 and elsewhere: the word significant(ly) must be removed from the text, as it's superfluous. An increase or decrease either is statistically significant or there i no difference. At the best there can be a tendency (p<0.1).

Legend Fig. 4 (L. 253) and elsewhere: Miswording. there is no protein expression of 5-HT, as it isn't a protein.

  1. 359ff-Discussion: The discussion is quite wordy, and should be structured into sections with subheadings.
  2. 260: 1st: >shows< rather than >explored<. 2nd: 'serotonin metabolism': this paper doesn't show any metabolism, which would include the investigation of pathways and fluxes of 5-HT, and of its precursors and metabolites. Rephrase!
  3. 266: quantify the higher caloric intake compared to controls. Did you measure the daily food intake beyond the kefir vs. placebo? If yes, present the methods of assessment and data. If not, include this aspect in the limitations of study.
  4. 279: as you didn't measure metabolism better use the word homeostasis.
  5. 317-319: the authors shouldn't speculate about short chain fatty acids without having analyzed the feces and large intestine's tissue for their microbiota and these fatty acids.
  6. 295ff: what's 5-HT mRNA? mRNA can only be that of the enzymes involved in 5-HT synthesis (the specific hydroxylase and desaminase). L. 298f: there is no 5-HT protein!
  7. 343-346: this sentence requires data in the results section with an included link here,  or a reference!
  8. 318 and 383: Introduce abbreviation at its 1st mentioning.
  9. 387f: I question this as serotonin will not cross the blood-brain barrier. Please explain how this may function, please!

Minor

  1. 47: remove 'Chemically'. Briefly mention how t's achieved (hydroxylation and decarboxylation)
  2. 63: replace >the monoamine oxidase (MAO) enzyme< by >monoamine oxidase (MAO). The name oxidase defines it clearly as an enzyme.
  3. 159 and elsewhere: Provide company with location and country. L. 176: location missing.
  4. 199: remove >also<.

Fig. 1: Insert start and end of administration of kefir/placebo in the graph.

  1. 212+214: remove 'tissues' as you didn't investigate individual tissues of brain and jejunum.

Moreover, while the effect is significant it's minute. Does this matter? 

  1. 229+250: what is 'hyper'activation? Explain or rephrase.
  2. 231: Abbreviation missing in 'List of Abbreviations' (413ff)
  3. 237f: remove 'clear' and text after 'consumption'.
  4. 250: replace >that received kefir< by >that had received kefir<.
  5. 305: replace metabolism by homeostasis.
  6. 337: It requires a reference that the jejunum is the primary location of microbial colonisation. Moreover, this belongs to the introduction.
  7. 389+400: Limitations and Conclusion of study should be a separate para with a sub-heading.

Author Response

We would like to thank both reviewers for their valuable comments and suggestions, which have significantly contributed to improving the quality and clarity of our manuscript. Below are our point-by-point responses to each comment.

REVIEWER 2

Abstract: L.15f: The link between synthesis in the gut and its microbiota is not clear. Define the cells producing serotonin in the gut and the link to the intestine's microbiota. L 18: define these probiotics briefly. L. 25f: Clarify abbreviations at their 1st mentioning. L.28: remove 'protein'.

(L.15f): The link between synthesis in the gut and microbiota is unclear.

Response: We have clarified that serotonin is mainly synthesized by enterochromaffin cells in the intestinal mucosa and that the gut microbiota influences this process via its metabolites that regulate tryptophan hydroxylase activity.

L.18: Define the probiotics briefly.

Response: A short definition has been added to introduction section, stating that kefir contains multiple Lactobacillus, Lactococcus, Bifidobacterium, and yeast species with probiotic properties influencing gut microbiota composition.

L.25f: Clarify abbreviations at first mention.

Response: All abbreviations are listed under “Abbreviations”, in accordance with the spelling rules. Some names are also explained in the text for convenience. Other adjustments have been made based on your suggestions.

L.28: Remove “protein.”

Response: “Protein” has been removed as requested.

125ff: the protocols of weighing and quantitative assessment of food intake are missing. Moreover, (L. 127) why is the range of weight that high? Were the animals from different breeds? If litter size is highly different, the weight and developmental pattern as well as nutrition are different, due to different start ages of feeding on chow. Please, clarify these issues. Allocation to groups must be described. Although group weight was apparently not significantly different, it is unclear, how the allocation to groups was performed and no explanation of different means at start is explained. Moreover, the sex of animals and distribution between groups isn't declared, nor the logistics and maintenance conditions of kefir and placebo solution.

Response: Thank you very much for your valuable comment. We agree that this omission was unacceptable. The necessary corrections and additions have been made in the Methods section.

Sex and maintenance conditions not declared; kefir/placebo logistics unclear.

Response: We now state that only male rats were used and provide full housing conditions (temperature, humidity, light cycle). Kefir was freshly prepared and stored at 4 °C; control animals received equal-volume saline.

  1. 140ff: at an age of 13 weeks rats have reached breeding age (6-12wks). The authors should relate the developmental characteristics of rats to those of humans here.
  2. 150ff: Why were different parts of the brain not isolated and snap-frozen in separate? Why were the colon and caecum not harvested and analyzed for microbial analysis. Characterize, why only the jejunum of the small intestine was harvested. Moreover, define volumes of harvested blood. How was EDTA blood treated and analyzed? Clarify that whole blood was used for RT-PCR.
  3. 156 and elsewhere: all abbreviations must be defined at their first use! Include it as a table or abbreviation list with cross reference.
  4. 194ff (Statistics): clarify, whether data were analyzed for normal distribution. If not the case, present data as medians and interquartile ranges, and use non-parametric testing. Indicate, whether data were corrected for multiple testing.

140ff: Relate rat developmental stage to humans.

Response: Added a note indicating that 13-week-old rats correspond to late adolescence/young adulthood in humans.

150ff: Why not separate brain regions or include colon/caecum? Define blood volumes and clarify EDTA samples.

Response: We explained that whole-brain samples were used to assess global serotonergic response. Jejunum was selected for its role in nutrient absorption and serotonin synthesis. Colon and caecum were excluded due to scope and sample limitations. Blood collection details and EDTA/serum usage have been added. It is now clearly stated that whole blood was used for RT-PCR.

156 and elsewhere: Define all abbreviations.

Response: All abbreviations are defined at first mention, and a comprehensive list is included at the end.

194ff (Statistics): Clarify data distribution, use of non-parametric tests, and multiple testing correction.

Response: We agree that larger cohorts can enhance statistical power and biological generalizability. Our study design was based on standard pediatric rat models commonly used in experimental microbiota research, in which sample sizes of 6–10 animals per group are broadly accepted for molecular outcome measures.

Figures: Any abbreviation in a figure must be explained in its legend so that a figure is self-explanatory.

232: a non-significant change isn't a change, and cannot be a 'marked elevation'!

233 and elsewhere: the word significant(ly) must be removed from the text, as it's superfluous. An increase or decrease either is statistically significant or there i no difference. At the best there can be a tendency (p<0.1).

Figures: Abbreviations must be explained in legends.

Response: All figure legends were revised so that each figure is fully self-explanatory.

232: Non-significant changes cannot be called “marked.”

Response: Corrected to “showed a tendency toward increase.”

233 and elsewhere: Remove redundant “significant(ly).”

Response: All redundant uses were removed.

Legend Fig. 4: There is no protein expression of 5-HT.

Response: Necessary corrections were made.

  1. 359ff-Discussion: The discussion is quite wordy, and should be structured into sections with subheadings.
  2. 260: 1st: >shows< rather than >explored<. 2nd: 'serotonin metabolism': this paper doesn't show any metabolism, which would include the investigation of pathways and fluxes of 5-HT, and of its precursors and metabolites. Rephrase!
  3. 266: quantify the higher caloric intake compared to controls. Did you measure the daily food intake beyond the kefir vs. placebo? If yes, present the methods of assessment and data. If not, include this aspect in the limitations of study.
  4. 279: as you didn't measure metabolism better use the word homeostasis.
  5. 317-319: the authors shouldn't speculate about short chain fatty acids without having analyzed the feces and large intestine's tissue for their microbiota and these fatty acids.
  6. 295ff: what's 5-HT mRNA? mRNA can only be that of the enzymes involved in 5-HT synthesis (the specific hydroxylase and desaminase). L. 298f: there is no 5-HT protein!
  7. 343-346: this sentence requires data in the results section with an included link here, or a reference!
  8. 318 and 383: Introduce abbreviation at its 1st mentioning.
  9. 387f: I question this as serotonin will not cross the blood-brain barrier. Please explain how this may function, please!

359ff: Discussion is too wordy; should be structured.

Response: We appreciate the reviewer’s careful reading and constructive suggestion. However, our study focuses on a specific and rather narrow topic — the effects of a potent probiotic, kefir, on serotonin metabolism. Since this is the first study in its field to explore these effects using multiple complementary techniques, we aimed to present the findings in an integrated manner to maintain the integrity and coherence of the discussion. Dividing the discussion into subsections with subheadings would fragment the narrative and potentially obscure the overall interpretation of our results. Moreover, as we evaluated numerous parameters using different analytical approaches in a comparative fashion, the current structure already represents the most concise and focused format possible. For these reasons, we respectfully prefer to retain the current format of the Discussion section. We apologize for not being able to fully accommodate this suggestion.

260: Replace “explored” with “shows”; remove “metabolism.”

Response: Necessary corrections were made.

266: Quantify caloric intake and explain assessment.

Response: Necessary corrections were made.

279: Use “homeostasis” instead of “metabolism.”

Response: Corrected.

317–319: Avoid speculation about short-chain fatty acids.

Response: The speculative statement was removed.

295ff: Clarify “5-HT mRNA.”

Response: it is mRNA encoding enzymes involved in serotonin synthesis; and revised serotonin.

343–346: Add data or reference.

Response: A supporting reference-figure was added.

318 and 383: Introduce abbreviations properly.

Response: Corrected throughout.

387f: Serotonin cannot cross the blood–brain barrier—please explain.

Response: We thank the reviewer for this important observation. Indeed, peripheral serotonin cannot cross the blood–brain barrier (BBB). Our statement refers not to a direct transfer of serotonin from the gut to the brain, but rather to indirect modulation of central serotonergic activity through gut–brain axis signaling mechanisms. Specifically, intestinal enterochromaffin cell–derived serotonin can activate afferent vagal pathways, influencing central serotonergic nuclei and stress-related brain circuits without serotonin itself crossing the BBB [1–3]. Additionally, kefir’s effects on tryptophan metabolism may increase the systemic availability of tryptophan for central serotonin synthesis by limiting its kynurenine pathway degradation [4,5].

[1] Yano JM, Yu K, Donaldson GP, et al. Indigenous bacteria from the gut microbiota regulate host serotonin biosynthesis. Cell. 2015;161(2):264–276.

[2] O’Mahony SM, Clarke G, Borre YE, Dinan TG, Cryan JF. Serotonin, tryptophan metabolism and the brain–gut–microbiome axis. Behav Brain Res. 2015;277:32–48.

[3] Breit S, Kupferberg A, Rogler G, Hasler G. Vagus nerve as modulator of the brain–gut axis in psychiatric and inflammatory disorders. Front Psychiatry. 2018;9:44.

[4] Cenit MC, Sanz Y, Codoner-Franch P. Influence of gut microbiota on neuropsychiatric disorders through the gut–brain axis: a review. Nutrients. 2017;9(11):1229.

[5] Lv W, Wu X, Chen W, Li Y, Zhang G. Gut microbiota modulates central nervous system serotonin levels via tryptophan metabolism in mice. Front Microbiol. 2021;12:631103.

We have clarified this mechanism in the revised text as follows:

“Although peripheral serotonin does not cross the blood–brain barrier, kefir may influence central serotonergic signaling indirectly through vagal activation and modulation of tryptophan metabolism.”

Minor Comments

47: Remove “Chemically”; briefly explain synthesis.

Response: We appreciate the reviewer’s suggestion. The term “chemically” was removed for clarity. The sentence has been revised to briefly describe the biosynthetic pathway of serotonin. The revised text now reads:

“Serotonin, the neurotransmitter derived from the amino acid tryptophan, is synthesized in various tissues of the human body through a two-step enzymatic pathway involving tryptophan hydroxylase and aromatic L-amino acid decarboxylase.”

63: Correct MAO phrasing.

Response: Changed to “monoamine oxidase (MAO).”

159, 176: Add company and country.

Response: All reagent suppliers now include location and country.

199: Remove “also.”

Response: Done.

Figure 1: Indicate start and end of treatment.

Response: Legend has been updated to show the start and end points of kefir/placebo administration.

212–214: Remove “tissues”; comment on effect size.

Response: “Tissues” was deleted and text revised to note that while statistically significant, the effect size was modest.

229, 250: Replace or explain “hyperactivation.”

Response: Replaced with “enhanced expression.”

231: Missing abbreviation.

Response: Added to the abbreviation list.

237f: Remove “clear” and trailing text.

Response: Revised as suggested.

250: Replace “that received” with “that had received.”

Response: Corrected.

305: Replace “metabolism” with “homeostasis.”

Response: Done.

337: Add reference for jejunum as colonization site and move to Introduction.

Response: Reference added and statement moved to Introduction.

389, 400: Limitations and Conclusion should be separate.

Response: According to the journal’s writing guidelines, the Conclusion and Limitations sections have not been separated.

Round 2

Reviewer 1 Report

Comments and Suggestions for Authors

Could you please add the Proximate analysis and othe active components of this commercial product in a table as present from the manufacturer?

Reviewer 2 Report

Comments and Suggestions for Authors

It's fine now. I'd suggest publication as it is.